# Fortification of Bread with Carob Extract: A Comprehensive Study on Dough Behavior and Product Quality

**DOI:** 10.3390/foods14101821

**Published:** 2025-05-20

**Authors:** Jana Zahorec, Dragana Šoronja-Simović, Jovana Petrović, Ivana Nikolić, Branimir Pavlić, Katarina Bijelić, Nemanja Bojanić, Aleksandar Fišteš

**Affiliations:** 1Faculty of Technology, University of Novi Sad, 21000 Novi Sad, Serbia; dragana@tf.uns.ac.rs (D.Š.-S.); jovana.petrovic@uns.ac.rs (J.P.); ivananikolic@unc.ac.rs (I.N.); bpavlic@uns.ac.rs (B.P.); bojanic@tf.uns.ac.rs (N.B.); fistes@uns.ac.rs (A.F.); 2Department of Pharmacy, Faculty of Medicine, University of Novi Sad, 21000 Novi Sad, Serbia; katarina.bijelic@mf.uns.ac.rs; 3Center for Medical and Pharmaceutical Investigations and Quality Control, Faculty of Medicine, University of Novi Sad, 21000 Novi Sad, Serbia

**Keywords:** bread fortification, carob extract, dough rheology, texture, sensory quality, polyphenols

## Abstract

The integration of functional ingredients into staple foods like bread offers a promising strategy for improving public health. Carob (*Ceratonia siliqua* L.) flour, rich in bioactive compounds, has potential as a functional additive. However, its incorporation into bread negatively affects dough behavior and product quality due to high levels of insoluble dietary fibers. This study investigates the use of carob extract (PCE) as a functional additive to enhance the nutritional and bioactive profile of bread while preserving its rheological behavior and sensory quality. PCE was obtained via microwave-assisted extraction and spray drying, and incorporated into bread formulations at 1%, 3%, and 5%. The addition of PCE reduced water absorption by 1.5% and increased dough stability three times. Dough resistance increased by 15%, while extensibility decreased by 5%. The viscoelastic properties of dough were preserved, as the storage modulus increased and Tan δ values remained stable. Changes in specific volume, crumb texture, crumb porosity, and bread color of produced bread with PCE were minimal; however, aroma, taste, and overall sensory quality were improved. Additionally, the incorporation of PCE resulted in a significant increase in total phenolic content and antioxidant activity, indicating an enhancement of the bread’s functional properties. These improvements were achieved without negatively affecting the dough rheology or bread quality parameters. Overall, the findings suggest that PCE can be a promising functional ingredient in bread formulations, contributing to both nutritional value and technological performance.

## 1. Introduction

The concept of functional food emerged from research into the health and physiological effects of various food ingredients, along with numerous scientific findings showing that a diet rich in foods containing active components is directly linked to a reduced risk of chronic noncommunicable diseases [1]. Functional food is defined as food that, in addition to its basic nutritional function, also offers health benefits due to the presence of nutraceuticals or bioactive agents [2]. The carriers of functionality in food ingredients or food products are the functional compounds found in its composition. Functional compounds include dietary fibers, polyphenols, carotenoids, fatty acids, plant sterols, prebiotics, probiotics, phytoestrogens, proteins, vitamins, and minerals [3]. In the field of functional food, products that represent basic staple foods have the greatest potential. Bread, in particular, provides significant opportunities for incorporating functional ingredients due to its frequent consumption, good taste, and wide consumer acceptance.

Earlier enrichment strategies primarily targeted specific nutrients, including folate [4], dietary fibers [5,6,7], vitamins, and minerals [8]. Subsequently, research interest has shifted toward the use of plant-based materials as nontraditional sources for enriching bakery products. More recently, carob (*Ceratonia siliqua* L.) has attracted increasing interest as a nontraditional ingredient with functional properties. Carob pulp is notably rich in insoluble dietary fiber and bioactive compounds such as polyphenols, flavonoids, vitamins, and minerals [9]. Carob pulp flour has been successfully integrated into various food products, including baked goods, beverages, and dairy products, which are increasingly present on the market in response to consumer demand for gluten-free, caffeine-free, and low-sugar alternatives [10,11]. Beyond its nutritional value, it has been used as a sugar substitute and a natural enhancer of antioxidant and sensory properties food products [12]. In particular, researchers have explored its incorporation into bread formulations, aiming to harness its functional benefits while adapting to consumer preferences and nutritional trends [13,14,15,16].

Despite its functional potential, incorporating carob flour into bread formulations presents technological challenges. Substitution of part of the wheat flour with carob flour alters the rheological properties of the dough, leading to reduced loaf volume and harder crumb structure [17]. The disruption of the characteristic physical properties of bread made with carob flour leads to a decrease in product acceptability among consumers [16,17]. Studies focusing on the impact of carob fibers have shown that the high content of insoluble fibers have significant effects on gluten network formation and dough structure [18]. Since functional food can only achieve the desired effects when products with modified compositions meet most consumer expectations [19], new approaches are needed to reduce or avoid the negative effects that arise during the development of functional bread.

In recent research, the focus of the scientific public has shifted toward the extraction and application of targeted bioactive compounds from plant materials. Polyphenol-rich extracts have emerged as promising alternatives in bread fortification, and incorporation of plant extracts into bread formulations has become a prominent trend in the development of new functional foods [20,21,22,23]. Numerous scientific studies highlight ongoing efforts to enhance the extraction of bioactive compounds from plant materials using advanced techniques, such as ultrasound-assisted extraction, pressurized liquid extraction, supercritical fluid extraction, and microwave-assisted extraction [24]. Additionally, various encapsulation methods are being developed to protect these bioactive compounds during subsequent processing and ensure their stability in the final product [25,26,27].

Unlike traditional methods of enriching bread, utilizing carob extracts, which are devoid of fibrous bulk, may offer a way to enhance functionality without compromising product quality [23]. However, the majority of current literature addresses the antioxidant potential of such extracts in final products, with limited focus on their influence during dough processing and baking. To the best of our knowledge, there are no published studies that comprehensively examine the effects of carob extracts in the bread production process. To bridge this knowledge gap, this study aims to examine the effects of carob extract on dough rheology, with a focus on the physical and sensory properties, nutritional improvements, and bioactive capacity of the bread. By addressing these aspects, the study provides significant insights into the functional application of carob extract in breadmaking, contributing to the development of nutritionally enhanced bakery products that meet both consumer health demands and technological feasibility.

## 2. Materials and Methods

### 2.1. Chemicals and Ingredients

Trolox (6-hydroxy-2,5,7,8-tetramethylchroman-2-carboxylic acid) was obtained from Sigma-Aldrich (Milano, Italy), while gallic acid was purchased from Sigma-Aldrich (St. Louis, MO, USA). ABTS (2,20-azino-bis-(-3-ethylbenzothiazoline-6-sulfonic acid) diammonium salt) was purchased from J&K Scientific GmbH (Pforzheim, Germany). Furthermore, Folin–Ciocalteu reagent, (±)-catechin, TPZT (2,4,6-tris(2-pyridil)-s triazine), DPPH• (1,1-diphenyl-2-picryl-hydrazyl-hydrate), potassium persulfate, iron(III)-chloride, and iron(II)-sulfatheptahydrate were obtained from Sigma-Aldrich (Steinheim, Germany). Additionally, the following standards of HPLC analytical grade were obtained from the same manufacturer: caffeic acid, chlorogenic acid, *trans*-cinnamic acid, *p*-coumaric acid, rosmarinic acid, gallic acid, ferulic acid, quercetin, rutin, and quercitrin.

The following commercially available ingredients were used in the breadmaking process:Wheat flour was supplied by the local mill (Kikindski mlin AD, Kikinda, Serbia) and had following chemical composition: moisture 11.8%, ash 0.55% d.w., protein 12.7% d.w., and wet gluten content 27.0%;Carob pulp flour originating from Croatia was purchased from local distributer Vega ADM (Senta, Serbia);Bakers’ yeast (Lesaffre, Budapest, Hungary);Salt (So product, Stara Pazova, Serbia).

### 2.2. Preparation of Carob Extract

To obtain a polyphenol-rich extract of carob pulp flour, the optimization of microwave-assisted extraction (MAE) was performed in our previous research and the process was described in detail [28]. Optimized carob extract (CE) with the highest yields of phenolic compounds and antioxidant activity was used in this research for the purpose of bread fortification. CE was obtained by mixing 2.5 g of plant material with 100 mL of 40% (*w*/*w*) ethanol in a round flask, which was placed into remodeled microwave oven (NN-E201W, Panasonic, Osaka, Japan) and extracted at 600 W for 35 min. By repeating the above process, a sufficient amount of CE was collected and prepared for drying. Spray-drying procedure used to convert liquid extract into dry form was described by Zahorec et al. [23] and was implemented in this work with some modifications. In short, CE was mixed with an aqueous solution of maltodextrin (DE 18.5) in a ratio of dry extract residue:maltodextrin = 70:30. The drying was performed in a pilot Anhydro spray dryer plant (APV Anhydro AS, Soborg, Denmark) with peristaltic pump (FH100 Series, Thermo Scientific, Waltham, MA, USA) and a flow rate of 1.36 L/h. Spray drying was performed at 160 °C (inlet temperature). The collected carob extract powder (PCE) was transferred to a glass vessel and kept at room temperature.

### 2.3. Characterization of Carob Extracts

#### 2.3.1. Chemical Profile

Total phenols content (TPC) in obtained carob extracts was determined by the Folin–Ciocalteu spectrophotometric method, while total flavonoids content (TFC) was determined by the colorimetric method, both described in detail in our previous research [28]. All measurements were performed in three replicates. The results for CE were expressed on a raw material basis, while those for PCE were expressed on a dry extract basis.

The method for identification and quantification of polyphenols in obtained carob extracts (CE and PCE) was previously described [29]. HPLC analysis was conducted using Agilent Technologies 1100 liquid chromatograph (Agilent Technologies, Santa Clara, CA, USA) equipped with a Nucleosil C18 column (250 mm, i.d. 4.6 mm, 5 μm particle size; Macherey Nagel, Düren, Germany). The chromatograms were monitored at three wavelengths according to the used chemical standards: at 280 nm (trans-cinnamic acid, gallic acid, and caffeic acid), at 330 nm (chlorogenic acid, ferulic acid, rosmarinic acid, p-coumaric acid, and quercetin), and at 350 nm (quercitrin and rutin). The quantity of phenolic compounds in CE was expressed in µg/g of raw material, while in PCE it was expressed in µg/g of dry extract.

#### 2.3.2. Antioxidant Activity

Antioxidant activity of carob extracts was determined by three in vitro methods: DPPH, ABTS, and FRAP. DPPH assay is based on the extracts’ capability to neutralize 1,1-diphenyl-2-pikrilhidrazil radicals (DPPH•); ABTS assay is used to assess the scavenging capacity of carob extracts against ABTS^+^ radical; and ferric ion reducing antioxidant power (FRAP) assay was applied to determine the reducing power of carob extracts towards Fe^3+^ ions. Mentioned tests were performed following the methodology described in previously published work [28]. All measurements were conducted in triplicate. The results for CE were reported on a raw material basis, while those for PCE were reported on a dry extract basis.

### 2.4. Rheological Properties of Dough

#### 2.4.1. Empirical Measurements

To assess the effect of PCE on dough properties, 1%, 3%, and 5% (g/100 g) PCE were added to the total flour mass, and the results were compared to the control dough (without PCE). Brabender farinograph (Duisburg, Germany) was used in the examination of mixing properties of dough samples and for determination of water absorption, development time, stability, and degree of softening according to the standard procedure [30]. Brabender extansograph (Duisburg, Germany) was used for the evaluation of extensional properties of dough samples and the following parameters were measured according to the official method [31]: energy, resistance to extension, extensibility, and resistance to extensibility ratio (R/E).

#### 2.4.2. Fundamental Measurements

Deeper insight into the influence of PCE on dough properties was obtained by applying fundamental rheological methods, namely dynamic oscillatory measurements and creep and recovery tests. The Haake Rheo Stress 600 rotary viscometer (Thermo Electron Cerporation, Karlsruhe, Germany) with the serrated set plate–plate sensor P35 Ti L (plate diameter 35 mm and gap 2 mm) was used in this research. The test samples were taken from the excess dough prepared for extensographic measurement (Section 2.3.1) and were placed between the plates of the rotary viscometer and allowed to rest for 3 min and was tempered to 30 °C. The measurement procedures, as well as the applied models for calculating the rheological parameters of the dough, were thoroughly described in our previously published work [32]. All measurements were performed in triplicate.

### 2.5. Laboratory Baking Test

Breadmaking process was visually presented in Figure 1. The composition of the control sample’s raw material was: wheat flour (900 g), yeast (2.5%, on flour basis), and salt (2%, on flour basis). Bread samples with added 1%, 3%, or 5% (on flour basis) powdered carob extract were labeled PCE1, PCE3, and PCE5, respectively.

The amount of water used for kneading corresponded to the water absorption determined by the farinograph for each sample. A high-speed spiral mixer (MS-6) was used for kneading. Bulk fermentation was conducted at 30 °C for 45 min, after which the dough was kneaded, and left for another 15 min in the fermentation chamber. The dough was the divided into three dough pieces of 450 g, rounded manually and left to relax for 10 min before the final shaping into loaves. Final proofing was conducted at 30 °C and 80% relative humidity for approximately 60 min. Finally, baking was done in a laboratory oven at 220 °C for 25–30 min.

### 2.6. Technological Quality of Bread

#### 2.6.1. Specific Volume

Specific volume (cm^3^/g) of bread samples was calculated as ratio of bread volume determined by the rapeseed displacement method [33] and its weight. Measurements were performed in triplicates.

#### 2.6.2. Crumb Structure by Digital Image Analysis

Digital image analysis was used for assessment of crumb structure of obtained bread samples [34]. Three central slices of the same sample were scanned and two crops of each scanned image (20 × 20 mm) were made. Recorded images were processed and analyzed using ImageJ 1.54d program (http://imagej.net/ij/, accessed on 20 March 2024). The smallest structure that can affect the result was defined by setting the sensitivity threshold at 0.1 mm. As a result of the analysis, data on the total number of pores and their surface area were obtained and the following parameters were calculated: pore density, average pore surface area, proportion of pore surface area, and proportion of pore walls surface area. Also, the pore size distribution (the proportion of pores expressed as a percentage of the total number of pores) was used to gain insight into crumb uniformity. According to the surface area, six pore size classes were selected: very small pores < 0.50 mm^2^; small pores = 0.50–0.99 mm^2^; medium pores = 1.00–1.99 mm^2^; large pores = 2.00–2.99 mm^2^; very large pores = 3.00–4.99 mm^2^; voids ≥ 5.00 mm^2^).

#### 2.6.3. Texture Profile Analysis

Texture Profile Analysis (TPA) test simulates the chewing process in the mouth and is commonly used for investigating of textural properties of bread crumb. In this work, bread samples were examined 24 h after baking using the TA.HDplus Texture Analyser (Stable Micro Systems, Godalming, UK). Bread loafs were cut into 18 mm thick slices, which were subjected to a weight load equivalent to 45% of the sample thickness. The initial testing speed was set at 10 mm/s, followed by a second speed of 5 mm/s [23]. Measurements were performed in five replicates and parameters hardness, springiness, cohesiveness, chewiness, and resilience were calculated from TPA plots using the Texture Exponent 32 software version 4.0.11.0 (Stable Micro Systems, Surrey, UK).

#### 2.6.4. Bread Color

Bread crumb and crust color was determined using a Minolta Chroma Meter CR-400 colorimeter, equipped with an 8 mm aperture on the measuring head (Konica Minolta Inc., Osaka, Japan), using the standard attachment CR-A33b. The color of raw materials was expressed in the CIE *L***a***b** color system [35] through coordinates: *L**—lightness of color, *a**—redness (+) or greenness (−), *b**—yellowness (+) or blueness (−).

#### 2.6.5. Sensory Properties

The assessment of sensory properties of bread samples was performed 24 h after baking according to the previously described method [17]. In short, a six-member panel of trained assessors evaluated external appearance, crumb structure, smell, and taste of bread samples with grades 1 to 5. The grades were multiplied with previously define importance factor and resulted in the number of points. Classification of bread samples was based on the sum of all evaluated quality attributes (total score), as follows: unacceptable (≤56.0), acceptable (56.1–67.0), good (67.1–78.0), very good (78.1–89.0), excellent (89.1–100.0).

Additionally, bread crumb quality was quantified by summing the numerical values assigned to descriptive grades for crumb elasticity and crumb pore structure. The total crumb quality score ranged from 0.0 to 7.0, following the method outlined by Kaluđerski and Filipović [36].

### 2.7. Chemical Profile of Bread

Total, soluble, and insoluble dietary fibers content was determined according to the official AOAC enzyme-gravimetric method No. 958.29 [37].

The procedures for determination of TPC and TFC, as well as the antioxidant activity by DPPH, FRAP, and ABTS assays, were the same as in Section 2.3.1. For determination of mentioned parameters in bread samples, bioactive compounds were first extracted from bread [17]. The extraction of compounds from bread was carried out in two ways: (1) from the crumb only—the crust was removed from the bread slice, and only the crumb was used for analysis; and (2) from the whole slice—including both crumb and crust, which were ground together and prepared for analysis. Homogenized sample of bread crumb/crumb + crust (3 g) was mixed with ethanol (15 mL, 80% *w*/*w*) in a glass cuvette. The mixture was tempered in a rotary thermostat for 3 h at room temperature. The content of the cuvette was then centrifuged at 4000 rpm for 10 min, the supernatant was separated by decantation and stored at 4 °C until analysis. All measurements were performed in three replicates and the results were expressed per 100 g of bread.

### 2.8. Statistical Analysis

For the estimation of the significance of the difference between mean values (*p* < 0.05) Duncan’s test was applied in the one-way ANOVA analysis. Software used for performing of statistical analysis was STATISTICA 14.0.0.15 (TIBCO Statistica™, Palo Alto, CA, USA).

## 3. Results

### 3.1. Characterization of CE and PCE

By the application of the microwave-assisted extraction (MAE) technique, a carob extract of the desired composition and high quality was obtained. The optimization of the MAE process has been previously discussed and verified [28]. However, the use of liquid extracts in the food industry is limited because they are sensitive to environmental factors. Exposure to light, oxygen, and moisture can cause degradation of their active compounds. In contrast, dry extracts offer several advantages: they are easier to store, more stable, and contain higher concentrations of active components [38]. Additionally, the extraction solvent for obtaining CE was 40% ethanol, which precludes its use in bread production in liquid form. Therefore, CE was converted into powder (PCE) using the spray drying technique. The characterization of PCE was based on the results of relevant biochemical indicators—TPC, TFC, and antioxidant activity—and were compared with those of CE (Table 1). Based on the presented results, the impact of drying on the mentioned quality parameters was assessed.

The results showed that the values of all the tested parameters are slightly lower in the PCE compared to CE, which was expected considering that drying involved the use of a carrier, which reduced the concentration of active principles in the dry extract. This reduction in the number of compounds with antioxidant activity consequently led to a decrease in antioxidant activity, as determined by all three in vitro tests (Table 1). However, the obtained results were considered satisfactory, since the differences in the results for TPC, TFC, and FRAP were not statistically significant (*p* > 0.05), while for DPPH and ABTS, differences were minimal.

Although photometric tests serve as an effective method for screening samples to determine the quantity of polyphenols and antioxidant components, they may not always accurately reflect the true phenol content due to potential cross-reactions with other reducing agents [39]. For this reason, HPLC analysis could offer a more precise assessment of the polyphenol composition in the extracts. The quantitative data obtained from HPLC analysis of both CE and PCE are presented in Table 1. The results revealed the presence of several compounds in the tested extracts, including chlorogenic acid, gallic acid, caffeic acid, rutin, quercitrin, and quercetin. Although quercetin was detected, its concentration in carob extracts was below the quantification limit (LOQ = 10.00 µg/g, Appendix A). The extracts showed gallic acid as the dominant compound, which aligns with findings from other studies [40,41]. Caffeic acid was the second most abundant compound, while other compounds were present in amounts less than 100 µg/g (Table 1).

The presented results show that for most of the identified compounds, there were no significant differences when comparing their concentrations in CE and PCE. However, statistically significant differences were observed for quercitrin and rutin, which could be due to the high temperature used during the spray drying process. Several scientific studies indicate that flavonols are particularly vulnerable to thermal degradation, even at temperatures below 160 °C, which was the drying temperature applied during CE drying [42,43].

### 3.2. Rheological Properties of Dough with PCE

#### 3.2.1. Empirical Rheological Parameters

There are no data in the literature on the effects of liquid or powdered carob extracts on bread dough properties. Therefore, in this work, every available method was applied in order to apprehend the changes occurring during the incorporation of PCE into the dough composition.

Based on the results of farinographic measurements presented in Table 2, it can be observed that the greatest changes in examined parameters were in the amount of water required for mixing and achieving constant dough consistency. The addition of 3% and 5% PCE caused a statistically significant decrease in water absorption by 1–1.5% compared to the control dough (*p* < 0.05). Given that PCE predominantly consists of polyphenolic compounds encapsulated in maltodextrin, the observed results are consistent with studies investigating the impact of maltodextrin addition on farinographic parameters of wheat dough [44]. The referenced study indicated that a higher proportion of maltodextrin with lower DE values exhibited a more pronounced effect in reducing water absorption. Furthermore, the reduced water absorption observed with higher amounts of PCE can be attributed to a decrease in the content of gluten, starch, and other flour components responsible for water binding in the total dough mass [45].

When observing the dough development, only the addition of 5% PCE significantly increased development time compared to the other samples, from about 2.0 to 9.0 min (Table 2). The corresponding farinograms revealed that in the PCE3 and PCE5 samples, two peaks were formed. For PCE3 the maximum consistency indicating the end of dough development occurred at the first peak, while for PCE5 it was observed at the second peak. This phenomenon suggests that the higher amount of PCE interferes with the formation of the gluten network. Based on literature, it is assumed that this effect results from interactions between polyphenols and gluten, which hinders dough development and prolongs the process [46,47]. Moreover, the presence of maltodextrin as a carrier of polyphenolic compounds during mixing may also contribute to the prolonged dough development [44,48].

A statistically significant effect on dough stability was observed when PCE was added to dough in amounts of 3% and 5%. An increase in mentioned parameter was recorded, reaching a 2- and 3-time higher value, respectively, compared to the control sample (Table 2). When observing the degree of dough softening, no clear trend was detected. However, in general, the addition of PCE lead to a slight but statistically significant decrease in the degree of softening (Table 2). The literature indicates that certain polyphenolic compounds, such as gallic or caffeic acid, which are present in PCE (Table 1), reduce dough tolerance during mixing [49]. However, this was not observed in the present study, and therefore, the positive impact on dough stability and softening could instead be attributed to maltodextrin [44].

By analyzing the results obtained by extensographic measurements (Table 2), clear trends were observed. Dough resistance was directly proportional, whereas extensibility was negatively correlated to the amount of PCE. The addition of PCE in the tested proportions resulted in a 5 to 15% increase in the dough resistance compared to the control, with the first significant change observed in the PCE3 sample. The extensibility values for dough samples with up to 3% PCE did not differ statistically (*p* < 0.05) from the control dough, while the PCE5 sample showed a statistically significant decrease in extensibility by 5%.

As a result of the occurred alterations in resistance and extensibility, changes in the values of energy and the resistance to extensibility ratio (R/E) were also observed. The energy of the dough increased in proportion to the amount of PCE added, with statistically significant differences observed across all samples (Table 2). The increase in energy was primarily due to the increase in dough resistance, as the extensibility remained relatively high regardless of the PCE proportion. However, the dough’s extensional properties with added PCE did not significantly differ from the control sample, as shown by the R/E results. Although an increase in R/E from 2.4 to 2.8 was observed, statistical significance between the values was not established. Dough that maintains a balanced ratio of R/E is considered to be of suitable quality for baking production [50]. The R/E values of dough with PCE were at the upper limit of the optimal range, further supporting the dough’s suitability for baking purposes.

#### 3.2.2. Fundamental Rheological Parameters

The assessment of the viscoelastic properties of dough with added PCE was carried out by determining the storage modulus (G’), loss modulus (G”), and rheological parameters from the creep and recovery tests.

The results showing the dependence of the viscoelastic moduli on frequency are presented in Figure 2a. It is evident that G’ was consistently higher than G” across all tested samples. Additionally, an increase in both viscoelastic moduli with rising frequency was observed in all dough samples, which is characteristic of dough made from wheat flour [51]. The high values of the moduli suggest a relatively high mobility of the gluten protein chains, enabling the dough to effectively adapt to the applied forces [52,53].

By fitting the experimental data of G’ against changes in frequency using a linear model (power law model), high values for the coefficient of determination (*R*^2^ ≥ 0.99) were obtained for all dough samples (Table 3). These results indicate a strong correlation between the power law model and the experimental data within the observed frequency range, confirming that the addition of PCE did not disrupt the viscoelastic nature of the system or the characteristic networked structure [54,55].

The presented results showed that with the increase in PCE proportion in the dough, G₀’ also increased. When 1% and 3% of PCE was added, G₀’ increased compared to the control sample, but no statistically significant difference was observed between the mentioned samples (Table 3). A significant increase in G₀’ was recorded in the PCE5 sample, both compared to the control and to samples with lower amounts of PCE (Table 3). These changes in the storage modulus suggest an enhancement in the dough’s elastic properties, indicating that the dough formed with PCE was stronger than the control sample [56] These findings align with the results of extensional measurements, particularly the observed increase in dough resistance as the PCE content increases (Table 2).

High values of the storage modulus alone do not necessarily indicate optimal elastic properties of the dough. To better understand the rheological behavior, Figure 2b illustrates the contribution of the viscoelastic moduli (G’ and G”) via the Tan δ parameter (G”/G’). This parameter reflects the system’s overall response to the application of nondestructive forces [57,58]. The results demonstrate that for all samples analyzed, the Tan δ value was <1, suggesting that elastic forces dominate over viscous forces. Furthermore, the addition of PCE in the range of 1% to 5% did not significantly change the Tan δ parameter when compared to the control sample, as confirmed by statistical analysis (*p* < 0.05). The consistency in the ratio of elastic to viscous forces across all samples indicates that the dough with added PCE maintains its viscoelastic nature.

The viscoelastic behavior of the dough over extended time periods, as well as its resistance to nondestructive forces, was assessed by analyzing the creep and recovery curves following the application of a constant stress [54]. The resulting characteristic curves are presented in Figure 3, and the parameters of the Burger model as a function of PCE amount in the dough, determined during both the creep and recovery phases, are summarized in Table 4.

The appearance of the creep curves for the tested samples further confirmed the typical viscoelastic behavior of the dough. High coefficients of determination for both the creep phase (*R*^2^ > 0.99) and the recovery phase (*R*^2^ > 0.98) confirmed a strong fit of the obtained curves with the Burger creep model (Table 4). The results indicate that the addition of PCE generally lead to an increase in the compliance parameters (J_0_, J_1_, and J_max_), with significant differences compared to the control sample observed in the dough with 3% and 5% PCE (Table 4). This increase in compliance values suggested that a higher proportion of PCE enhanced the dough’s ability to deform more easily under constant stress [59]. Furthermore, as PCE content increased, a slight decrease in Newtonian viscosity was observed, most notably at the highest PCE levels, with a significant difference from the control sample (*p* < 0.05). The reduction in η_0_, alongside the increase in J_max_, indicated a softening effect on the dough structure.

All tested dough samples containing PCE exhibited viscosity and adaptability values that were relatively similar to those of the control sample. Therefore, the results of the fundamental rheological measurements confirmed the findings from the empirical measurements, which indicated that the addition of PCE has a minimal impact on the dough’s rheological properties. Based on these findings, it can be expected that PCE will not influence the dough’s behavior negatively during bread production, nor will it significantly affect the quality of the final product.

### 3.3. Bread Quality with the Addition of PCE

#### 3.3.1. Physical and Textural Characteristics

By examining the specific volume, one of the key physical indicators of bread quality, it was observed that the addition and increase in the amount of PCE in the bread composition led to a slight decrease in the value of this parameter (Table 5). The largest decrease in specific volume relative to the control sample is approximately 10%, occurring at the addition of the maximum tested amount of PCE. Although the changes in the specific volume of bread with added PCE were not substantially large, they were statistically different from the control sample (*p* < 0.05). These results are in agreement with the empirical test findings—the minimal effect of PCE on R/E combined with its positive impact on dough energy, suggesting that even bread with the highest tested amount of PCE would still have a relatively large volume. The observed reduction in the volume of bread with PCE can be associated with the influence of certain polyphenols. Han and Koh [49] reported that phenolic acids, particularly caffeic acid, which is present in significant amount in PCE, can reduce the proportion of high-molecular-weight proteins and increase the proportion of soluble proteins, thus leading to a restructuring of the gluten network, and consequently, diminishing the baking quality of the flour.

An insight into the bread crumb structure, which is closely related to volume, was obtained from the results of digital image analysis of the crumb section of the tested samples (Table 5). The presented results clearly show that bread samples with the addition of PCE had a slightly lower pore density compared to the control sample, with no significant differences between the samples with varying amounts of PCE. Additionally, the average pore area of the observed samples was very similar, although a slight increase in this parameter was noted with the increase in PCE amounts from 1% to 3% and 5%. Furthermore, it is important to note that the PCE3 and PCE5 samples showed a higher average pore area compared to the control sample. The PCE1 sample had the lowest proportion of pore surface area, while for PCE5 sample this value was closest to that of the control sample. The objective evaluation of the number of pores in predefined size classes is shown in Figure 4. Based on the presented results, it is evident that the distribution of pore sizes in the examined samples was relatively uniform. In all bread samples, pores with a surface area of <0.05 mm^2^ were dominant, accounting for more than 60% of the total number of pores. Nevertheless, the addition of PCE caused a slight reduction in the number of very small pores compared to the control sample. Medium, large, and very large pores were present in less than 10% in all samples, while voids (>5.00 mm^2^) accounted for 10–15%. However, it is noticeable that the control sample had a lower proportion of pores in mentioned size classes compared to the samples with PCE. The most pronounced differences among samples were observed in the 0.50–0.99 mm^2^ and 1.00–4.99 mm^2^ size classes. In both cases, the PCE5 sample exhibited an almost identical proportion of pores as the control.

The observed effects were expected and confirmed the assumptions made by observing the resistance, extensibility, as well as the G’ and J_max_ in rheological measurements. Namely, it was predicted that the obtained dough samples would have good stretchability under gas pressure during fermentation, while also maintaining sufficient strength to retain produced gas. Since these properties are crucial for obtaining a high-quality bread, it was logical that the addition of PCE had a minimal negative impact on volume and porosity. The appearance of sliced samples of bread with 1%, 3%, and 5% PCE is shown in Figure 5, which illustrates mentioned observations.

The values of the bread texture parameters, determined using the Texture Profile Analysis (TPA) test, are summarized in Table 6 and clearly indicate that the addition of PCE up to 5% did not negatively affect bread texture. The highest hardness value was registered for the control (1.66 kg), while the hardness values for the samples with PCE, regardless of the amount used, were approximately 1.40 kg. Statistical analysis confirmed that there were no significant differences in bread hardness among the observed samples, which aligns with the fact that all samples containing PCE exhibited high volume and well-developed crumb structures—quality indicators known to positively and strongly correlate with mentioned texture parameter [60].

Analysis of the chewiness, which is directly related to hardness, revealed a statistically significant difference between the control and all PCE samples. The observed decrease in chewiness in breads with PCE compared to the control suggests that PCE contributed to the softening of the crumb. These findings are consistent with previous reports on the effect of maltodextrin on bread texture [45,61]. The results obtained for the remaining texture parameters confirmed that the texture of the PCE samples was comparable to that of the control. High values of springiness and cohesiveness indicate that the crumb of all bread samples had the ability to return to its original shape without major structural disruption, while low resilience values further support these observations.

Since the primary goal of incorporating PCE into bread was to produce a product with characteristics similar to those of conventional white bread, the effect of PCE on color parameters was evaluated using instrumental measurements. From the results presented in Table 7 it is evident that the addition of PCE caused a darkening of both the crust and the crumb compared to the control sample. However, statistically significant differences were only observed at the 3% addition level. Furthermore, increasing the PCE content from 3% to 5% did not result in a significant further reduction in bread lightness (*L** value).

The most prominent changes in color characteristics were recorded in the *a** parameter (redness), showing a clear trend of increasing values with higher PCE levels. Significant deviations in *a** values were detected even at the lowest tested concentration of PCE. In the crust, the *a** value increased approximately 2 to 2.5 times compared to the control (Table 7). In the crumb, the initially negative *a** value—indicating a green hue in the control sample—shifted to a positive value, corresponding to a red hue, with increasing PCE. The intensity of the red tone in the crumb significantly increased in proportion to the amount of PCE used in the bread formulation. No notable differences were observed in the *b** parameter (yellowness) between the crust and crumb of the control and samples containing PCE.

#### 3.3.2. Sensory Characteristics

Considering that even minor changes in the raw material composition can lead to alterations in sensory properties, the effect of the addition of PCE on the most significant sensory quality parameters of bread was further analyzed. According to the results (Table 8 and Appendix A) the addition of PCE had no effect on the external appearance of the bread. All samples with PCE, as well as the control sample, received the highest scores, meaning they obtained the maximum points for this parameter. However, noticeable differences were observed for the other parameters. The bread samples PCE3 and PCE5 had a better crumb structure compared to that of the control bread and PCE1 sample.

A deeper insight into the bread crumb structure was provided by evaluating crumb quality, which represents the sum of scores for crumb elasticity and crumb pore structure [62,63]. Analyzing the results for these parameters, it is evident that the effect of PCE on crumb elasticity was minimal (Figure 6). Crumb elasticity for the sample with 1% PCE was comparable to the control sample, while bread with 3% PCE exhibited slightly better elasticity, and bread with 5% PCE showed slightly worse elasticity in comparison to the control. These results align with the trend observed in instrumental springiness measurements using the TPA test (Table 6). Changes in crumb quality values are mainly due to the effect of PCE on crumb pore structure, with a positive trend observed as the amount of PCE increased.

When observing the smell and taste of the bread, the results clearly show that the addition of PCE had a strong positive effect, as these sensory parameters for the control sample were rated with the lowest number of points (Table 8). Even the addition of the lowest tested amount of PCE increased the points for smell from 14.0 to 18.0, and for the taste from 21.0 to 27.0. Increasing the PCE content to 3% and 5% resulted in bread with the maximum score for mentioned parameters. Based on the evaluated sensory parameters, the enhancement in bread quality with the addition of PCE resulted in a higher total score compared to the control sample. For the PCE3 and PCE5, this also corresponded to a higher quality category (Table 8). These findings indicate that higher levels of carob extract can be incorporated into bread formulations without negatively affecting sensory quality. On the contrary, they may enhance consumer-perceived product attributes. Therefore, the results of this study may serve as a useful basis for developing formulation guidelines for functional bakery products enriched with plant extracts, particularly carob, supporting both nutritional improvement and sensory acceptance.

#### 3.3.3. Nutritional and Chemical Properties

Since the addition of PCE did not negatively impact the physical and sensory properties of the bread, the sample with the highest tested PCE concentration was considered optimal and was further evaluated for its content of dietary fibers, total phenolic content (TPC), total flavonoid content (TFC), and antioxidant activity. A comparison of the dietary fiber content between the PCE5 sample and the control bread (Table 9) revealed a significant increase in dietary fibers, particularly soluble fibers, which increased from 0.69 g/100 g in the control to 2.0 g/100 g in PCE5. The chemical composition of carob pulp flour is primarily composed of total carbohydrates (up to 86%), of which approximately 47% constitute total dietary fibers [10]. The largest proportion consists of insoluble fibers such as cellulose, hemicellulose, and lignin; however, a significant number of soluble fibers, including pectins, mucilages, and inulin, is also present [64]. Based on the presented results, it is assumed that, in addition to bioactive compounds, fibers are also extracted from carob flour during the extraction process. The results also demonstrated that the use of carob extract, instead of carob flour, in bread formulation allows for the exclusion of insoluble dietary fibers (Table 9), which are known to negatively affect bread quality [17].

The results obtained for the content of polyphenols and antioxidant activity in the observed bread samples confirmed that the addition of PCE resulted in bread with increased levels of total phenols and enhanced antioxidant activity (Table 9). TPC in the crumb of PCE5 was about 4 times higher compared to control, and 5 times higher when considering TPC in the entire bread, including the crust. Similarly, TFC in bread with added PCE was higher in comparison to the control bread, with higher amounts of active compounds detected when the analysis also included the bread crust. However, it was observed that the analysis of TFC in the examined bread samples was affected by certain interferences, which compromised the clarity of the results—similar to what was noted in our previous research on bread enriched with carob flour [17]. Additionally, the PCE5 sample exhibited higher antioxidant activity compared to the control, with the extent of the observed changes depending on the type of antioxidant test applied. Table 9 shows that PCE5 demonstrated a 3-, 5-, and 12-time higher antioxidant activity than the control sample, as measured by the ABTS, DPPH, and FRAP tests, respectively. A similar trend was observed when comparing antioxidant activity in the bread crumb versus the whole bread, consistent with previously discussed parameters. This enhancement is attributed to the presence of Maillard reaction products formed during baking, which contribute to the increased antioxidant activity in the crust and crumb of the bread [17]. These findings align with recent studies indicating that the incorporation of carob extract into bread formulations significantly elevates the total phenolic content and antioxidant capacity, both in the crumb and the whole bread, compared to control samples [23].

## 4. Conclusions

Bread with PCE was fortified with polyphenols exhibiting pronounced antioxidant activity, which, combined with the increased content of soluble dietary fibers, contributes to the development of a new functional product. By adding carob extract to the bread formulation, the impact of structural components of carob flour on the physical and sensory quality parameters of the product was avoided. As a result, a product similar to basic types of bread was achieved. Bread with PCE maintained unaltered sensory properties, which can ensure broader acceptability among consumers. Its higher acceptability could potentially lead to increased consumption of bioactive compounds in the daily diet compared to standard wheat bread, thus offering various health benefits.

## Figures and Tables

**Figure 1 foods-14-01821-f001:**
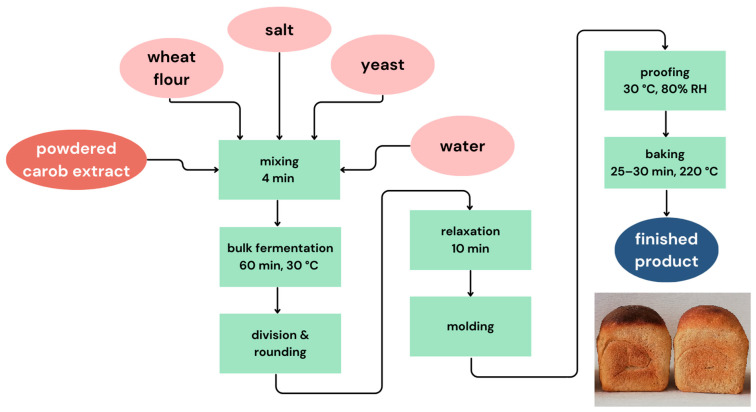
Flowchart of the bread production process.

**Figure 2 foods-14-01821-f002:**
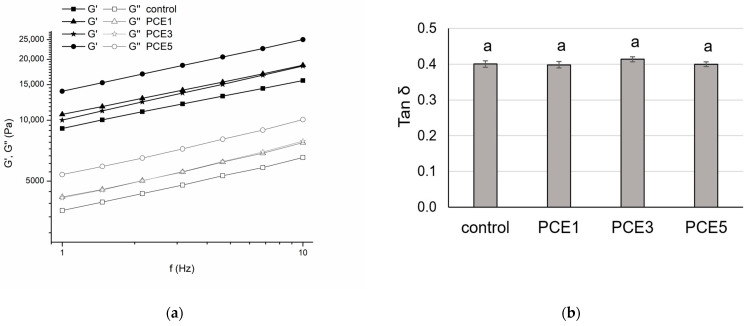
The effect of the addition of 1%, 3%, and 5% powdered carob extract (PCE) on: (**a**) storage modulus (G′) and loss modulus (G″) and (**b**) Tan δ (G″/G′). Values on the bars marked with different letters are statistically significantly different (*p* < 0.05).

**Figure 3 foods-14-01821-f003:**
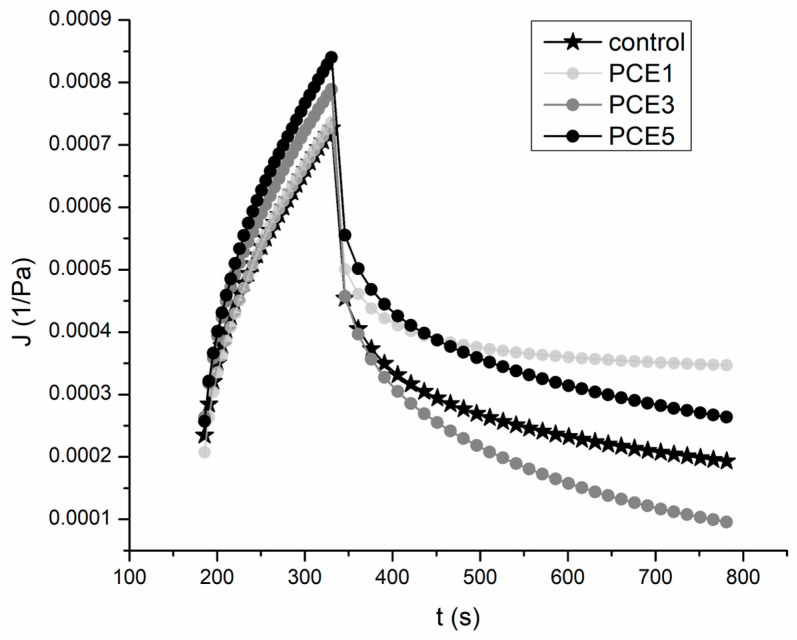
The effect of the addition of 1%, 3%, and 5% powdered carob extract (PCE) on the appearance of the creep and recovery curves.

**Figure 4 foods-14-01821-f004:**
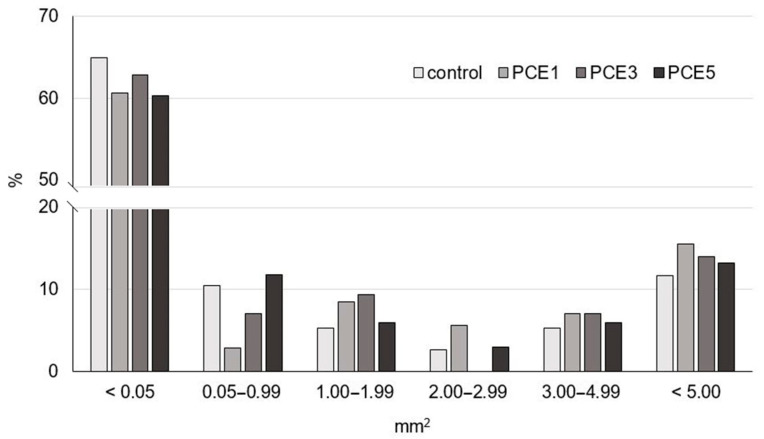
Effect of powdered carob extract (PCE) addition on the pore size distribution for the selected pore size classes.

**Figure 5 foods-14-01821-f005:**
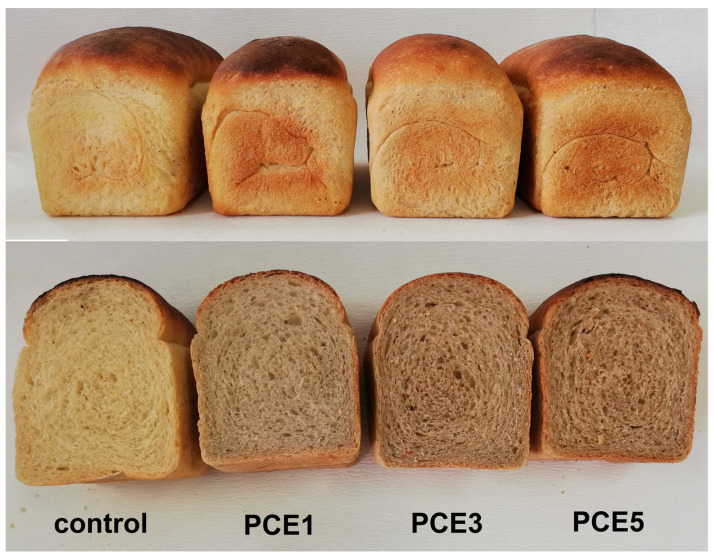
Effect of powdered carob extract (PCE) addition on the external appearance and the appearance of the bread crumb.

**Figure 6 foods-14-01821-f006:**
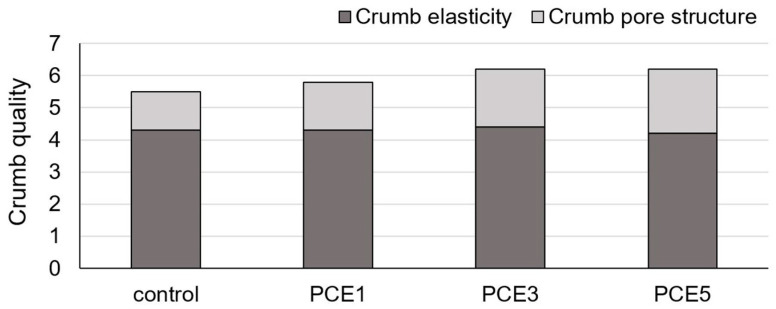
Effect of powdered carob extract (PCE) addition on the crumb quality parameters (crumb elasticity—max 4.5; crumb pore structure—max 3.5; crumb quality—max 7.0).

**Table 1 foods-14-01821-t001:** Polyphenols content, antioxidant activity, and individual compounds identified by HPLC in liquid and powdered carob extract.

Parameter	CE ^1^	PCE
Polyphenols content		
TPC (mg GAE/100 g)	1609.92 ± 56.15 ^a^	1545.78 ± 74.42 ^a^
TFC (mg CE/100 g)	271.92 ± 5.73 ^a^	267.72 ± 6.25 ^a^
Antioxidant activity		
DPPH (μM TE/g)	99.02 ± 0.60 ^a^	92.34 ± 0.15 ^b^
FRAP (μM Fe^2+^/g)	50.45 ± 5.36 ^a^	46.16 ± 2.66 ^a^
ABTS (μM TE/g)	110.55 ± 0.97 ^a^	102.90 ± 0.03 ^b^
Compound (µg/g)		
*trans*-cinnamic acid	<LOD ^2^	<LOD
caffeic acid	259.3 ± 13.0 ^a^	240.2 ± 12.0 ^a^
chlorogenic acid	83.6 ± 4.2 ^a^	73.9 ± 3.7 ^a^
*p*-coumaric acid	<LOD	<LOD
ferulic acid	<LOD	<LOD
gallic acid	2371.9 ± 355.8 ^a^	2475.5 ± 371.3 ^a^
rosmarinic acid	<LOD	<LOD
quercetin	<LOQ	<LOQ
quercitrin	153.5 ± 7.7 ^a^	74.9 ± 3.7 ^b^
rutin	233.1 ± 18.6 ^a^	94.9 ± 7.6 ^b^

^1^ CE—carob extract, PCE—powdered carob extract. ^2^ LOD—limit of detection; LOQ—limit of quantification. Mean ± standard deviation (n = 3); values in rows marked with different letters are statistically significantly different (*p* < 0.05).

**Table 2 foods-14-01821-t002:** The influence of powdered carob extract on empirical rheological parameters.

Sample ^1^	Water Absorption (%)	Development Time (min)	Stability (min)	Degree of Softening (FU)
control	56.03 ± 0.08 ^c^	2.0 ± 0.2 ^a^	0.5 ± 0.3 ^a^	45.0 ± 0.5 ^d^
PCE1	56.17 ± 0.11 ^c^	2.3 ± 0.3 ^a^	0.5 ± 0.2 ^a^	35.0 ± 1.5 ^b^
PCE3	55.20 ± 0.16 ^b^	2.0 ± 0.1 ^a^	1.0 ± 0.1 ^b^	40.0 ± 1.0 ^c^
PCE5	54.67 ± 0.12 ^a^	9.0 ± 0.2 ^b^	1.5 ± 0.3 ^c^	30.0 ± 1.0 ^a^
	**Energy (cm^2^)**	**Resistance (EU)**	**Extensibility (mm)**	**R/E ^2^**
control	124.5 ± 0.84 ^a^	400 ± 10 ^a^	167.5 ± 3.0 ^b^	2.39 ± 0.17 ^a^
PCE1	136.5 ± 1.06 ^b^	420 ± 15 ^ab^	165.5 ± 4.2 ^b^	2.54 ± 0.29 ^a^
PCE3	140.0 ± 0.50 ^c^	450 ± 20 ^b^	160.0 ± 4.5 ^ab^	2.81 ± 0.42 ^a^
PCE5	143.8 ± 0.97 ^d^	455 ± 10 ^b^	158.5 ± 2.3 ^a^	2.87 ± 0.21 ^a^

^1^ PCE1, PCE3, PCE5—1%, 3%, and 5% of powdered carob extract. ^2^ R/E—resistance to extensibility ratio. Results represent mean values of repeated measurements ± SD (n = 3); values in columns marked with different letters are statistically significantly different (*p* < 0.05).

**Table 3 foods-14-01821-t003:** Parameters of the power law equation describing the dependence of the storage modulus (G’) on frequency.

Sample *	G_0_′ (kPa)	n	*R* ^2^
control	9.17 ± 0.08 ^a^	0.234 ± 0.002 ^a^	0.9999
PCE1	10.15 ± 0.04 ^b^	0.243 ± 0.001 ^b^	0.9999
PCE3	10.02 ± 0.11 ^b^	0.266 ± 0.001 ^d^	1.000
PCE5	13.90 ± 0.05 ^c^	0.255 ± 0.004 ^c^	1.000

* PCE1, PCE3, PCE5—1%, 3%, and 5% of powdered carob extract; G_0_′—the intercept of the line on the x-axis; n—the corresponding slope of the line; *R*^2^—the coefficient of determination. Results represent mean values of repeated measurements ± SD (n = 3); values in columns marked with different letters are statistically significantly different (*p* < 0.05).

**Table 4 foods-14-01821-t004:** Dependence of the Burger model parameters during the creep and recovery phases on the amount of powdered carob extract (PCE).

Parameter ^2^	Control	PCE1 ^1^	PCE3	PCE5
Creep phase				
J_0_ (10^−4^·Pa^−1^)	2.35 ± 0.05 ^b^	2.07 ± 0.13 ^a^	2.63 ± 0.02 ^d^	2.57 ± 0.01 ^c^
η_0_ (10^5^·Pas)	4.55 ± 0.20 ^b^	4.49 ± 0.27 ^b^	4.19 ± 0.21 ^ab^	3.94 ± 0.16 ^a^
J_1_ (10^−4^·Pa^−1^)	2.40 ± 0.05 ^a^	2.43 ± 0.11 ^a^	2.60 ± 0.06 ^b^	2.77 ± 0.03 ^b^
J_max_ (10^−4^·Pa^−1^)	7.27 ± 0.03 ^a^	7.36 ± 0.32 ^a^	7.89 ± 0.09 ^b^	8.40 ± 0.13 ^c^
λ (s)	143.8 ^a^	143.8 ^a^	143.8 ^a^	143.8 ^a^
*R* ^2^	0.9973	0.9974	0.9967	0.9967
Recovery phase				
J_0_ (10^−4^·Pa^−1^)	4.54 ± 0.14 ^a^	5.01 ± 0.08 ^b^	4.62 ± 0.11 ^a^	5.55 ± 0.02 ^c^
η_0_ (10^5^·Pas)	6.37 ± 0.02 ^c^	2.25 ± 0.16 ^a^	2.46 ± 0.20 ^a^	2.96 ± 0.11 ^b^
J_1_ (10^−4^·Pa^−1^)	0.64 ± 0.01 ^a^	1.15 ± 0.12 ^c^	1.05 ± 0.01 ^c^	0.87 ± 0.10 ^b^
λ (s)	339.6 ^a^	339.8 ^a^	339.6 ^a^	339.6 ^a^
*R* ^2^	0.9019	0.9915	0.9872	0.9888

^1^ PCE1, PCE3, PCE5—1%, 3%, and 5% of powdered carob extract. ^2^ J_0_—instantaneous compliance, J_1_—retarded elastic compliance or viscoelastic compliance, η_0_—Newtonian viscosity, λ—retardation time, J_max_—maximum creep compliance obtained at the end of the creep phase, *R*^2^—the coefficient of determination. Results represent mean values of repeated measurements ± SD (n = 3); values in rows marked with different letters are statistically significantly different (*p* < 0.05).

**Table 5 foods-14-01821-t005:** The effect of powdered carob extract (PCE) amount on the specific volume and crumb pore characteristics of bread.

Sample *	Specific Volume (cm^3^/g)	Pore Density (1/cm^2^)	Average Pore Surface Area (mm^2^)	Proportion of Pore Surface Area (%)	Proportion of Pore Walls Surface Area (%)
control	3.04 ± 0.12 ^b^	19.3	2.46	47.31	52.69
PCE1	2.87 ± 0.15 ^ab^	17.5	2.44	45.35	54.65
PCE3	2.74 ± 0.05 ^a^	17.8	2.57	46.72	53.29
PCE5	2.74 ± 0.01 ^a^	17.0	2.76	47.00	53.00

* PCE1, PCE3, PCE5—1%, 3%, and 5% of powdered carob extract. Mean ± SD (n = 3); values in column marked with different letters are statistically significantly different (*p* < 0.05).

**Table 6 foods-14-01821-t006:** The effect of powdered carob extract (PCE) amount on the textural properties of bread.

Sample *	Hardness (kg)	Springiness	Resilience	Cohesiveness	Chewiness (kg)
control	1.660 ± 0.465 ^a^	0.708 ± 0.091 ^a^	0.383 ± 0.027 ^a^	0.749 ± 0.062 ^a^	2.376 ± 0.319 ^b^
PCE1	1.444 ± 0.529 ^a^	0.712 ± 0.093 ^a^	0.377 ± 0.029 ^a^	0.767 ± 0.078 ^a^	1.135 ± 0.374 ^a^
PCE3	1.441 ± 0.243 ^a^	0.720 ± 0.089 ^a^	0.387 ± 0.028 ^a^	0.745 ± 0.058 ^a^	1.288 ± 0.691 ^a^
PCE5	1.435 ± 0.280 ^a^	0.697 ± 0.094 ^a^	0.377 ± 0.021 ^a^	0.764 ± 0.059 ^a^	1.483 ± 0.235 ^a^

* PCE1, PCE3, PCE5—1%, 3%, and 5% of powdered carob extract. Results represent mean values of repeated measurements ± SD (n = 5); values in columns marked with different letters are statistically significantly different (*p* < 0.05).

**Table 7 foods-14-01821-t007:** The effect of powdered carob extract (PCE) amount on the crust and crumb color of bread.

Sample *	Crust Color		
	*L**	*a**	*b**
control	70.42 ± 1.11 ^b^	3.26 ± 0.97 ^a^	28.96 ± 0.96 ^a^
PCE1	64.78 ± 1.10 ^ab^	6.26 ± 0.85 ^bc^	30.02 ± 0.71 ^a^
PCE3	59.86 ± 1.81 ^a^	8.28 ± 1.59 ^c^	29.57 ± 1.71 ^a^
PCE5	57.71 ± 5.15 ^a^	8.73 ± 1.10 ^c^	29.98 ± 2.09 ^a^
	**Crumb color**		
	***L****	***a****	***b****
control	65.82 ± 1.97 ^b^	−1.29 ± 0.07 ^a^	16.06 ± 0.66 ^b^
PCE1	61.83 ± 2.18 ^ab^	0.40 ± 0.12 ^b^	15.95 ± 0.70 ^b^
PCE3	58.79 ± 1.81 ^a^	2.25 ± 0.21 ^c^	16.04 ± 1.01 ^b^
PCE5	57.74 ± 1.23 ^a^	3.14 ± 0.06 ^d^	17.27 ± 0.45 ^b^

* PCE1, PCE3, PCE5—1%, 3%, and 5% of powdered carob extract; *L**—lightness of color, *a**—redness (+) or greenness (−), *b**—yellowness (+) or blueness (−). Results represent mean values of repeated measurements ± SD (n = 3); values in columns marked with different letters are statistically significantly different (*p* < 0.05).

**Table 8 foods-14-01821-t008:** Sensory analysis results of bread with added powdered carob extract (PCE).

Sample *	External Appearence	Crumb Structure	Smell	Taste	Total Score	Quality Category
control	20.0	24.0	14.0	21.0	79.0	Very good
PCE1	20.0	24.0	18.0	27.0	89.0	Very good
PCE3	20.0	27.0	20.0	30.0	97.0	Excellent
PCE5	20.0	27.0	20.0	30.0	97.0	Excellent

* PCE1, PCE3, PCE5—1%, 3%, and 5% of powdered carob extract; External appearance and smell—max 20.0; Crumb structure and taste—max 30.0; total score—max 100.0.

**Table 9 foods-14-01821-t009:** Dietary fibers, polyphenols content, and antioxidant activity of bread samples.

Parameter	Control		PCE5 *	
Dietary fibers content				
Total dietary fibers (g/100 g)	1.18 ± 0.12 ^a^		2.65 ± 0.09 ^b^	
Insoluble dietary fibers (g/100 g)	0.49 ± 0.17 ^a^		0.65 ± 0.22 ^a^	
Soluble dietary fibers (g/100 g)	0.69 ± 0.18 ^a^		2.00 ± 0.18 ^b^	
	**control** **(crumb)**	**control** **(crumb + crust)**	**PCE5** **(crumb)**	**PCE5** **(crumb + crust)**
Polyphenols content				
TPC (mg GAE/100 g)	11.52 ± 1.45 ^a^	12.67 ± 0.88 ^a^	46.37 ± 3.37 ^b^	60.53 ± 3.77 ^c^
TFC (mg CE/100 g)	38.29 ± 1.76 ^a^	45.45 ± 1.09 ^b^	50.17 ± 0.58 ^c^	58.11 ± 1.80 ^d^
Antioxidant activity				
DPPH (μM TE/100 g)	32.92 ± 4.32 ^a^	40.85 ± 1.14 ^b^	164.15 ± 4.85 ^c^	181.48 ± 5.75 ^d^
FRAP (μM Fe^2+^/100 g)	154.95 ± 6.69 ^a^	157.66 ± 4.71 ^a^	440.89 ± 4.45 ^b^	533.32 ± 18.54 ^c^
ABTS (μM TE/100 g)	17.09 ± 9.08 ^a^	27.40 ± 9.10 ^a^	214.85 ± 12.28 ^b^	225.94 ± 6.21 ^b^

* PCE5—bread with 5% of powdered carob extract. Mean ± standard deviation (n = 3); values in rows marked with different letters are statistically significantly different (*p* < 0.05).

## Data Availability

The original contributions presented in the study are included in the article/Appendix A. Further inquiries can be directed to the corresponding author.

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
