# Peer review of "Fortification of Bread with Carob Extract: A Comprehensive Study on Dough Behavior and Product Quality"

_foods, 2025, doi:10.3390/foods14101821_

Round 1

Reviewer 1 Report

Comments and Suggestions for Authors

foods-3634668

Title: Fortification of Bread with Carob Extract: A Comprehensive Study on Dough Behavior and Product Quality

This study aimed to investigate the role of carob extract on dough behavior and product quality of bread. The results showed that the addition of carob extract (PCE) reduced water absorption and extensibility, while increasing dough stability and resistance. In addition, aroma, taste, and overall sensory quality were improved, and total phenolic content and antioxidant activity were significantly elevated by PCE addition . In all, the current manuscript contains valuable information, but it is necessary to consider the following issues.

Here are some specific comments.

  1. What advantage dose carobhave compared to other legumes?
  2. What component does the PCE contain besides polyphenols?
  3. In the part of 3.3.3. Nutritional and chemical properties, the PCE5 sample was chosen for further analyzing the nutritional and chemical properties, why are other groups not considered.
  4. When researcher conducted the experiment of the antioxidant activityon crumb and crust, were the samples fully resolved completely.
  5. In line 174, 30℃ >-- 30℃, it should be standardized to 30℃ throughout the manuscript.
  6. The results showed that the sample with the highest tested PCE concentrationhave higher content of dietary fibers, the potential mechanism behind increase should be further analyzed.

Author Response

Comment 1: Title: Fortification of Bread with Carob Extract: A Comprehensive Study on Dough Behavior and Product Quality

This study aimed to investigate the role of carob extract on dough behavior and product quality of bread. The results showed that the addition of carob extract (PCE) reduced water absorption and extensibility, while increasing dough stability and resistance. In addition, aroma, taste, and overall sensory quality were improved, and total phenolic content and antioxidant activity were significantly elevated by PCE addition. In all, the current manuscript contains valuable information, but it is necessary to consider the following issues.

Response 1: On behalf of all authors, we would like to thank the reviewer for the positive feedback and recognition of the study’s valuable findings. We have done our best to address all raised concerns and provide further clarification in the manuscript. All the changes regarding Reviewers’ comments were highlighted in the manuscript with yellow color.

Comment 2: Here are some specific comments. 1. What advantage dose carob have compared to other legumes?

Response 2: Thank you for your question. Carob offers several advantages compared to other legumes. Compared to legumes like soy or peas, carob has a lower protein and fat content, while the content of dietary fiber is higher compared to other legumes. Also, a wider range of minerals was found in carob compared to pea and lentil. Additionally, carob pulp stands out among legumes due to its notably higher content of D-pinitol. What makes carob pulp especially interesting for research is the fact that it is considered a by-product or in some cases waste in the industrial processing of carob fruit, and its utilization for human nutrition is highly significant. However, this section highlighting carob in regards to other legumes was erased from the Manuscript based on the requests from other Reviewers.

Comment 3: 2. What component does the PCE contain besides polyphenols?

Response 3: Thank you for your question. In this study, we primarily focused on the polyphenolic content of the PCE, since they were the targeted compounds of the extraction. Literature suggests that extracts from legumes could also contains sugars, and possibly proteins, however this was not investigated in carob extracts. Based on the data presented in our manuscript regarding the nutritional composition of the bread, we can confirm that PCE contains insoluble fibers and some minerals (K, Ca, Mg, Cu - these results are under consideration elsewhere).

Comment 4: 3. In the part of 3.3.3. Nutritional and chemical properties, the PCE5 sample was chosen for further analyzing the nutritional and chemical properties, why are other groups not considered.

Response 4: Thank you for your comment. The PCE5 sample was selected for further analysis due to its optimal concentration, which provided the best balance between technological performance and nutritional enhancement. Other concentrations were not considered in this part of the study, as our primary goal was to focus on the most promising formulation.

Comment 5: 4. When researcher conducted the experiment of the antioxidant activity on crumb and crust, were the samples fully resolved completely.

Response 5: Thank you for your comment. When conducting the antioxidant activity experiment, bread slices were taken for both the “crumb” and the “crumb + crust” samples. For the crumb analysis, the crust was discarded, and only the crumb was used for the analysis. In the case of whole bread slice analysis (crumb + crust), the entire slice was ground and homogenized before the antioxidant activity analysis. This clarification has been added to the Methods section 2.7.

Comment 6: 5. In line 174, 30℃ >-- 30℃, it should be standardized to 30℃ throughout the manuscript.

Response 6: Thank you for pointing that out. We have standardized the temperature symbol to 30℃ throughout the manuscript as suggested. Additionally, we followed the formatting guidelines, which recommend writing the number separately from the unit (e.g., 30 ℃) throughout the manuscript.

Comment 7: 6. The results showed that the sample with the highest tested PCE concentration have higher content of dietary fibers, the potential mechanism behind increase should be further analyzed.

Response 7: Thank you for your comment. We have elaborated the discussion regarding the increase in dietary fiber content in the sample with the highest tested PCE concentration. This has been addressed in section 3.3.3. Nutritional and chemical properties of the manuscript.

Reviewer 2 Report

Comments and Suggestions for Authors

The manuscript explores the effects of a polyphenol rich carob extract (obtained by microwave-assisted extraction technique) on dough rheology and physical, sensory, nutritional, and bioactive properties of the bread. The manuscript is well-written and easy to follow, with a clear structure and attention to detail.

Some comments:

L 78: “bioactive capacity of the bread”: I suggest to replace capacity with properties or other one

L 148: “extansograph“ – correct the name of the instrument

L 229: change enzyme to enzymatic

L 280: keep only the value for LOQ. I suggest to include the values of LOD and LOQ, in the footnote from Table 1.

Table 9: the statistical letters are missing for TPC, TFC and antioxidant activity.

Sensorial analysis: a panel with 6 members in sensory evaluation it seems quite low. Moreover, the initial sensory results (the notes) are not presented to allow to identify if there were differences between the answers of the 6 evaluators.

Author Response

Comment 1: The manuscript explores the effects of a polyphenol rich carob extract (obtained by microwave-assisted extraction technique) on dough rheology and physical, sensory, nutritional, and bioactive properties of the bread. The manuscript is well-written and easy to follow, with a clear structure and attention to detail.

Response 1: On behalf of the authors, I would like to express gratitude for your positive and thoughtful feedback. We appreciate your recognition of the manuscript’s clarity, structure, and attention to detail. Your comments are encouraging, and we are grateful for your time and effort in reviewing our work.

Comment 2: Some comments: L 78: “bioactive capacity of the bread”: I suggest to replace capacity with properties or other one

Response 2: It has been done. All the changes made according to the Reviewers’ comments are highlighted in turquoise color in the revised version of the manuscript.

Comment 3: L 148: “extansograph“ – correct the name of the instrument

Response 3: It has been done.

Comment 4: L 229: change enzyme to enzymatic

Response 4: It has been done.

Comment 5: L 280: keep only the value for LOQ. I suggest to include the values of LOD and LOQ, in the footnote from Table 1.

Response 5: L 300 - It has been done accordingly. Since LOD and LOQ are different for every compound, the values for each were presented in Supplementary material in Table S1.

Comment 6: Table 9: the statistical letters are missing for TPC, TFC and antioxidant activity.

Response 6: We would like to thank the reviewer for pointing out this oversight in Table 9. We have now added the corresponding letters to ensure clarity and proper statistical interpretation.

Comment 7: Sensorial analysis: a panel with 6 members in sensory evaluation it seems quite low. Moreover, the initial sensory results (the notes) are not presented to allow to identify if there were differences between the answers of the 6 evaluators.

Response 7: We acknowledge the importance of using a larger panel size. For sensory evaluation in this study, descriptive sensory analysis performed by trained experts was used. Authors can confirm that for this method, 6-10 evaluators are sufficient. While smaller panel sizes are generally less statistically powerful, they can still provide valuable preliminary data. We plan to expand the panel size and conduct further research with larger groups in future studies, since the results indicate that even larger amount of PCE can be added into bread formulation and not deteriorate products properties. Furthermore, we have included the data of initial sensory results in the Supplementary material, Table S2. We hope this addresses the reviewer’s concerns.

Reviewer 3 Report

Comments and Suggestions for Authors

The primary research question investigates how fortifying wheat bread with various concentrations of carob extract affects dough rheology, baking performance, and final product quality, including nutritional and sensory properties. Bread fortification with plant-derived ingredients is a timely research area, driven by consumer demand for functional foods with added health benefits. This study specifically addresses a gap by evaluating carob extract—a rich source of phenolics and antioxidants—rather than the more commonly studied carob flour or fiber. This contributes to diversifying natural fortification agents and examines their impact on both the technological and nutritional profiles of bread. While the methodology is generally sound, the following improvements are recommended: Include a detailed compositional analysis of the carob extract (e.g., phenolic profile) to correlate specific compounds with observed effects. Perform statistical tests beyond ANOVA for multivariate data interpretation (e.g., PCA or PLS for correlating rheological and antioxidant data). Evaluate shelf-life stability and bioaccessibility of phenolics post-baking to assess health relevance. Use larger sample sizes for sensory panels to improve statistical power.

The references are appropriate and cover relevant studies on carob, bread fortification, rheological testing, and antioxidant evaluation. However, a few more recent studies (within the last 2 years) could strengthen the literature review and contextualize the novelty of using carob extract specifically . The tables are clear, with relevant statistical annotations (e.g., superscript letters indicating significance). Figures such as rheological profiles and sensory scores are informative but could benefit from enhanced clarity (e.g., larger font sizes, clearer legends). Including visual bread images (crumb and crust) for each sample could further support sensory and texture interpretations.

Summary: The manuscript titled "Fortification of Bread with Carob Extract: A Comprehensive Study on Dough Behavior and Product Quality" presents a timely and original investigation into the use of carob extract as a functional ingredient in bread formulation. The study is well-structured and addresses a relevant gap by exploring the effects of bioactive-rich carob extract—rather than the more commonly used carob flour—on dough rheology, baking quality, antioxidant properties, and sensory acceptability.

The authors employ appropriate analytical techniques, and the conclusions are supported by the data. Notably, the study provides valuable insights into the trade-offs between nutritional enhancement and technological performance. However, some methodological aspects could be strengthened, such as providing a detailed compositional analysis of the extract, using advanced statistical tools, and assessing phenolic bioaccessibility post-baking. The figures and tables are generally clear, although minor improvements in graphical clarity are recommended.

Overall, the manuscript offers a meaningful contribution to the field of functional bakery products and is suitable for publication after minor revisions.

Author Response

Comment 1: The primary research question investigates how fortifying wheat bread with various concentrations of carob extract affects dough rheology, baking performance, and final product quality, including nutritional and sensory properties. Bread fortification with plant-derived ingredients is a timely research area, driven by consumer demand for functional foods with added health benefits. This study specifically addresses a gap by evaluating carob extract—a rich source of phenolics and antioxidants—rather than the more commonly studied carob flour or fiber. This contributes to diversifying natural fortification agents and examines their impact on both the technological and nutritional profiles of bread.

Response 1: We thank the reviewer for the positive comments and for highlighting the relevance of our research focus.

Comment 2: While the methodology is generally sound, the following improvements are recommended:

Include a detailed compositional analysis of the carob extract (e.g., phenolic profile) to correlate specific compounds with observed effects.

Perform statistical tests beyond ANOVA for multivariate data interpretation (e.g., PCA or PLS for correlating rheological and antioxidant data).

Evaluate shelf-life stability and bioaccessibility of phenolics post-baking to assess health relevance.

Use larger sample sizes for sensory panels to improve statistical power.

Response 2:

We thank the reviewer for the valuable suggestions. Regarding the detailed compositional analysis, we would like to clarify that a comprehensive breakdown of the carob extract, including the phenolic profile, has already been provided in Table 1 – section 3.1 ("Characterization of CE and PCE"), where the polyphenols profile was determined using HPLC.

We thank the reviewer for the valuable suggestions and insightful guidance, which would strengthen the quality of our work. We thank the reviewer for the suggestion to perform additional statistical tests. However, for this study, we found that ANOVA was sufficient to draw the necessary conclusions. While we did not use more advanced methods in this research, we will certainly consider incorporating them in future studies to provide deeper insights.

As for the evaluation of bioaccessibility of phenolics post-baking, we agree that this would be a valuable addition to our study. However, we currently lack the necessary equipment to perform these analyses. If future collaborations provide the opportunity, we would be happy to address this aspect in subsequent research.

We acknowledge the importance of using a larger panel size to improve statistical power. For sensory evaluation in this study, descriptive sensory analysis performed by trained experts was used. Authors can confirm that for this method, 6-10 evaluators are sufficient. While smaller panel sizes are generally less statistically powerful, they can still provide valuable preliminary data. We plan to expand the panel size and conduct further research with larger groups in future studies, since the results indicate that even larger amount of PCE can be added into bread formulation and not deteriorate products properties. We hope this addresses the reviewer’s concerns.

Comment 3: The references are appropriate and cover relevant studies on carob, bread fortification, rheological testing, and antioxidant evaluation. However, a few more recent studies (within the last 2 years) could strengthen the literature review and contextualize the novelty of using carob extract specifically. The tables are clear, with relevant statistical annotations (e.g., superscript letters indicating significance). Figures such as rheological profiles and sensory scores are informative but could benefit from enhanced clarity (e.g., larger font sizes, clearer legends). Including visual bread images (crumb and crust) for each sample could further support sensory and texture interpretations.

Response 3: We thank the reviewer for the helpful suggestions. We have included more recent studies within the last two years to further strengthen the literature review. Additionally, we have improved the quality of the figures to the best of our abilities, following the reviewer's comments on enhancing clarity. We have also added an image of the bread samples (bread crust appearance Figure 5a).

Comment 4: Summary: The manuscript titled "Fortification of Bread with Carob Extract: A Comprehensive Study on Dough Behavior and Product Quality" presents a timely and original investigation into the use of carob extract as a functional ingredient in bread formulation. The study is well-structured and addresses a relevant gap by exploring the effects of bioactive-rich carob extract—rather than the more commonly used carob flour—on dough rheology, baking quality, antioxidant properties, and sensory acceptability.

Response 4: We thank the reviewer for the positive feedback and for recognizing the originality and relevance of using carob extract in bread formulation.

Comment 5: The authors employ appropriate analytical techniques, and the conclusions are supported by the data. Notably, the study provides valuable insights into the trade-offs between nutritional enhancement and technological performance. However, some methodological aspects could be strengthened, such as providing a detailed compositional analysis of the extract, using advanced statistical tools, and assessing phenolic bioaccessibility post-baking. The figures and tables are generally clear, although minor improvements in graphical clarity are recommended.

Response 5: We appreciate the reviewer’s feedback, which has certainly helped enhance the quality of our work. To the best of our knowledge, we have addressed all the raised concerns and made the necessary adjustments. All the changes according to the Reviewers’ comments are highlighted with green color in the manuscript.

Comment 6: Overall, the manuscript offers a meaningful contribution to the field of functional bakery products and is suitable for publication after minor revisions.

Response 6: On behalf of our research team, I would like to thank the reviewer for their thorough review of our manuscript, for the positive assessment and for recognizing the potential contribution of our manuscript.

Reviewer 4 Report

Comments and Suggestions for Authors

Reviewers comments
The manuscript deals with the “Fortification of Bread with Carob Extract: A Comprehensive 2 Study on Dough Behavior and Product Quality”. This article considers investigating the effects of carob extract on dough rheology, with a comprehensive focus on the physical and sensory properties, nutritional improvements, and bioactive capacity of the bread.
Overall, the topic is interesting and catches the audience’s attention, but it needs to improve according to the suggested comments.
The abstract is well-structured and written. A couple of amendments will be useful for readers. 
[1] lines 26-28: Additionally, total phenolic content and antioxidant activity were significantly higher, confirming the potential of PCE to improve the functional value of bread without compromising technological performance. Please rearrange these sentences and add a proper conclusion in the abstract.
[2] The introduction is well-written, with clear background information provided on the important characteristics. The introduction needs more information. A brief description of the significance of this work should be added to the last paragraph of the introduction, which would be helpful for readers. 
[3] The lacking part of the introduction is a clear statement of the research gap or need for the current study. While the introduction mentions several studies but does not explicitly state the specific gap or problem the current study aims to address. In addition, there is no direct link between the previous studies and the objectives of the current study. Add recent references.
[4] Although the manuscript is well-written, a few grammatical errors and typos exist, which should be corrected at the revision stage.
[5] Lines 593-597: Table 9 shows that PCE5 demonstrated 3, 5, and 12 times higher antioxidant activity than the control sample, as measured by the ABTS, DPPH, and FRAP tests, respectively. A similar trend was observed when comparing antioxidant activity in the bread crumb versus the whole bread, consistent with previously discussed parameters. Please add the reason behind with latest references.
[6] Figure 1. Flowchart of the bread production process. Please add product photo. 
[7] lines 562-565 Increasing the PCE content to 3% and 5% resulted in bread with the maximum score for mentioned parameters. Based on the evaluated sensory parameters, the enhancement in bread quality with the addition of PCE resulted in a higher total score compared to the control sample. The information presented in this article should help provide the basis for such guidelines.      

Comments on the Quality of English Language

Minor English editing is needed.

Author Response

Comment 1: The manuscript deals with the “Fortification of Bread with Carob Extract: A Comprehensive 2 Study on Dough Behavior and Product Quality”. This article considers investigating the effects of carob extract on dough rheology, with a comprehensive focus on the physical and sensory properties, nutritional improvements, and bioactive capacity of the bread.

Response 1: On behalf of the author I would like the Reviewer for the comments and for taking time in reviewing our manuscript.

Comment 2: Overall, the topic is interesting and catches the audience’s attention, but it needs to improve according to the suggested comments.

Response 2: Thank you for your comments and for acknowledging the comprehensive scope of our manuscript. We tried to address all raised points and the changes made in the manuscript according to the Reviewers’ comments are highlighted in pink color.

Comment 3: The abstract is well-structured and written. A couple of amendments will be useful for readers.

Response 3: We thank the Reviewer for the positive feedback regarding the structure and clarity of the abstract.

Comment 4: [1] lines 26-28: Additionally, total phenolic content and antioxidant activity were significantly higher, confirming the potential of PCE to improve the functional value of bread without compromising technological performance. Please rearrange these sentences and add a proper conclusion in the abstract.

Response 4: We thank the reviewer for the suggestion. The final part of the abstract has been revised to improve clarity and to include a more conclusive summary of the findings, as recommended.

Comment 5: [2] The introduction is well-written, with clear background information provided on the important characteristics. The introduction needs more information. A brief description of the significance of this work should be added to the last paragraph of the introduction, which would be helpful for readers.

Response 5: We thank the reviewer for the suggestion. The final paragraph of the Introduction has been revised to better highlight the significance of the study.

Comment 6: [3] The lacking part of the introduction is a clear statement of the research gap or need for the current study. While the introduction mentions several studies but does not explicitly state the specific gap or problem the current study aims to address. In addition, there is no direct link between the previous studies and the objectives of the current study. Add recent references.

Response 6: We thank the reviewer for the comment. In response, we have revised parts of the Introduction to more explicitly state the research gap addressed by the current study. We also made sure to draw a clearer connection between the previous studies and the objectives of our research and have incorporated more recent references where applicable.

Comment 7: [4] Although the manuscript is well-written, a few grammatical errors and typos exist, which should be corrected at the revision stage.

Response 7: It has been done.

Comment 8: [5] Lines 593-597: Table 9 shows that PCE5 demonstrated 3, 5, and 12 times higher antioxidant activity than the control sample, as measured by the ABTS, DPPH, and FRAP tests, respectively. A similar trend was observed when comparing antioxidant activity in the bread crumb versus the whole bread, consistent with previously discussed parameters. Please add the reason behind with latest references.

Response 8: According to the Reviewers’ comment the explanation along with recent references has been added to clarify the observed antioxidant activity differences.

Comment 9: [6] Figure 1. Flowchart of the bread production process. Please add product photo.

Response 9: The product photo was added to the flowchart of the bread production.

Comment 10: [7] lines 562-565 Increasing the PCE content to 3% and 5% resulted in bread with the maximum score for mentioned parameters. Based on the evaluated sensory parameters, the enhancement in bread quality with the addition of PCE resulted in a higher total score compared to the control sample. The information presented in this article should help provide the basis for such guidelines.

Response 10: We thank the reviewer for the suggestion. This point has been addressed and included in the discussion.

Reviewer 5 Report

Comments and Suggestions for Authors

This manuscript investigates the incorporation of processed carob extract into bread formulations to enhance nutritional and bioactive properties while evaluating its effects on dough behavior and product quality.

A number of methodological and interpretative aspects merit further clarification and are outlined in the following comments:

  1. No HPLC chromatograms are presented in the manuscript; could the authors provide these chromatograms in the supplementary material to substantiate the reported analytical results?
  2. The manuscript lacks a discussion on the current use, functional properties, and market presence of carob in food products. It would strengthen the context and relevance of the study to address whether carob-derived products are already commercially available, how widely they are used, and their popularity or acceptance in the food industry.
  3. The inclusion of compounds in the table that are below the limit of detection (LOD) or limit of quantification (LOQ) is unclear; could the authors clarify the purpose of including these values and how they contribute to the interpretation of the results?

A significant proportion of the references cited (49 out of 65) are more than five years old, which is unusual for a study aiming to contribute to an evolving field; the inclusion of more recent literature would help ensure the manuscript reflects the current state of research and supports its relevance.

Author Response

Comment 1: This manuscript investigates the incorporation of processed carob extract into bread formulations to enhance nutritional and bioactive properties while evaluating its effects on dough behavior and product quality.

Response 1: On behalf of authors I would like to thank Reviewer for taking time to review our manuscript.

Comment 2: A number of methodological and interpretative aspects merit further clarification and are outlined in the following comments:

Response 2: We will try to address all the raised concerns in order to elevate the quality of this work. All changes in the manuscript according to the Reviewers’ comments are highlighted in grey color.

Comment 3: 1. No HPLC chromatograms are presented in the manuscript; could the authors provide these chromatograms in the supplementary material to substantiate the reported analytical results?

Response 3: According to the Reviewers’ comment we have presented HPLC chromatograms in the Supplementary material (Figure S1).

Comment 4: 2. The manuscript lacks a discussion on the current use, functional properties, and market presence of carob in food products. It would strengthen the context and relevance of the study to address whether carob-derived products are already commercially available, how widely they are used, and their popularity or acceptance in the food industry.

Response 4: We appreciate the Reviewer’s insightful comment. We have expanded the Introduction to include a discussion on the requested subject.

Comment 5: 3. The inclusion of compounds in the table that are below the limit of detection (LOD) or limit of quantification (LOQ) is unclear; could the authors clarify the purpose of including these values and how they contribute to the interpretation of the results?

Response 5: We thank the reviewer for this pertinent observation. The values below LOD or LOQ were included in the table to provide a comprehensive overview of all analytes monitored during the analysis, including those not detected or present at trace levels. This approach allows readers to clearly distinguish between compounds that were not detected at all, those detected but not reliably quantified, and those present in quantifiable concentrations. Including these data points also supports transparency in reporting and highlights the sensitivity limitations of the analytical method used. In the revised manuscript, we have added Table S1 into Supplementary material to clarify the notation and explain the interpretive value of

Comment 6: A significant proportion of the references cited (49 out of 65) are more than five years old, which is unusual for a study aiming to contribute to an evolving field; the inclusion of more recent literature would help ensure the manuscript reflects the current state of research and supports its relevance.

Response 6: We thank the reviewer for this valuable observation. We acknowledge that a significant portion of the references cited are older than five years, and we understand the importance of grounding our work in the most recent scientific developments. However, we would like to clarify that many of the older references have been included due to their foundational nature, particularly in relation to carob composition and analytical techniques, which remain relevant and have not been substantially updated in recent literature. It is also important to note that the novelty of our work lies not in the extraction of carob compounds—an area that has indeed been well covered in recent years—but in the investigation of how carob extract affects the quality of bread. To the best of our knowledge, very few studies have explored this specific application, which limits the availability of recent references directly related to our research focus. Nevertheless, in response to the reviewer’s suggestion, we have revised the manuscript and incorporated several recent references (from the past five years) where applicable in the Introduction to ensure the manuscript reflects the current state of research and supports its relevance to the evolving field of functional bakery products.

Round 2

Reviewer 5 Report

Comments and Suggestions for Authors

Dear Authors,

Thank you for your comprehensive and thoughtful responses to my comments. I have reviewed the revised manuscript along with your point-by-point replies and the accompanying supplementary materials and I am satisfied that all my concerns have been adequately addressed. From my perspective, no further revisions are required.